

# Early-life intestinal microbiome in *Trachemys scripta elegans* analyzed using 16S rRNA sequencing

Qin Peng[1], Yahui Chen[1], Li Ding[1], Zimiao Zhao[1], Peiyu Yan[1], Kenneth B. Storey[2], Haitao Shi[1] and Meiling Hong[1]

[1] Ministry of Education Key Laboratory for Ecology of Tropical Islands, College of Life Sciences, Hainan Normal University, Haikou, Hainan, China
[2] Department of Biology, Carleton University, Ottawa, Canada

## ABSTRACT

During the early-life period, the hatchlings of red-eared slider turtles (*Trachemys scripta elegans*) rely on their own post-hatching internal yolk for several days before beginning to feed. The gut microbiome is critical for the adaptation of organisms to new environments, but, to date, how the microbiome taxa are assembled during early life of the turtle is unknown. In this study, the intestinal microbiome of red-eared slider hatchlings (fed on commercial particle food) was systematically analyzed at four different growth stages (0 d, 10 d, 20 d, 30 d) by a high-throughput sequencing approach. Results showed that the dominant phyla were Firmicutes (58.23%) and Proteobacteria (41.42%) at 0-day, Firmicutes (92.94%) at 10-day, Firmicutes (67.08%) and Bacteroidetes (27.17%) at 20-day, and Firmicutes (56.46%), Bacteroidetes (22.55%) and Proteobacteria (20.66%) at 30-day post-hatching. Members of the Bacteroidaceae family were absent in 0-day and 10-day turtles, but dominated in 20-day and 30-day turtles. The abundance of *Clostridium* also showed the highest value in 10-day turtles. The richness of the intestinal microbiomes was lower at 0-day and 30-day than that at 10-day and 20-day, while the diversity was higher at 10-day and 30-day than that at 0-day and 20-day. The results endowed the turtles with an ability to enhance their tolerance to the environment.

## INTRODUCTION

The microbial community in the gastrointestinal tract has a major role to play in several physiology processes of the host, such as maintaining intestinal microecological balance, promoting host health, and providing nutrients (*Chung et al., 2012*; *Kahrstrom, Pariente & Weiss, 2016*; *Mohd Shaufi et al., 2015*; *Yamashiro, 2017*). In addition, the gut microbiome is also associated with gut disease, feed conversion, parasite colonization, and immune system activity of the host (*Li et al., 2017*; *Singh et al., 2012*). Gut microbes can prevent the overgrowth of gut pathogens by building a natural barrier, called 'colonization resistance' that can inhibit the growth of pathogenic bacteria by occupying the same ecological niche (*Buffie & Pamer, 2013*; *Ducarmon et al., 2019*; *Scott et al., 2015*). Early promotion of nutrient metabolism and innate immune response depend upon the bacterial species that

Corresponding author
Meiling Hong,
mlhong@hainnu.edu.cn

colonize the digestive tract. Therefore, research on the microbiota of animals is gaining popularity in order to understand the relationships between host health, immunity, and disease resistance.

The red-eared slider turtle (*Trachemys scripta elegans*), with a native range in the southeastern U.S.A. and northeastern Mexico, is now one of the most successful invasive species in many regions of the world (*Mali et al., 2015*). It has been reported to outcompete several native freshwater turtles in Europe and Asia, including *Mauremys sinensis*, *Mauremys leprosa*, and *Mauremys reevesii* (*Nishizawa et al., 2014*; *Polo-Cavia, Lopez & Martin, 2012*). Several studies have been conducted in an effort to understand the invasion mechanism of the red-eared slider, including examining their feeding kinematics (*Nishizawa et al., 2014*), home range (*Ma et al., 2013*), food snatch ability, and hunger endurance ability (*Zhao et al., 2013*). Compared to native species, red-eared sliders exhibit unique advantages including holding an ability to live in brackish water that allows them to disperse along coastlines (*Yang & Shi, 2014*), greater food-competing ability, higher tolerance of starvation, higher reproductive capacity, and a greater ability to adapt to different environment by exploit new food sources or supplies (*Ma et al., 2013*; *Zhao et al., 2013*). Several of these advantages might be associated with the gut microbiome, that can influence food digestion and nutrient absorption, but little is known about the adaptive capacity of the intestinal microbiome of red-eared sliders. The existing studies on this species are limited to culture-dependent phenotypic and biochemical characterization of the gut microorganisms, and are not sufficient for systematically understanding the gut microecosystem of red-eared sliders (*Gioia-Di Chiacchio et al., 2014*; *Gaertner et al., 2008*).

High-throughput analysis of bacteria diversity and abundance can provide a greater understanding of the gut microecosystem of an animal host and potentially contribute to understanding the invasive potential of different species (*Ahasan et al., 2017*; *Han et al., 2015*; *Kumar et al., 2015*; *McLaughlin, Cochran & Dowd, 2015*; *Qin et al., 2010*; *Sergeant et al., 2014*; *Zeng et al., 2015*). The community of the gut microbiome varies at different growth stages (*Arizza et al., 2019*; *Burgos, Ray & Arias, 2018*; *Campos et al., 2018*; *Dulski, Zakes & Ciesielski, 2018*; *Huang et al., 2014*). The onset of feeding is a critical stage in the development of animals (*Sarasquete, Polo & YúFera, 1995*). During the early-life period, turtles rely on their own post-hatching yolks (absorbed into the coelomic cavity) for several days, and then begin to feed. By selecting for bacteria associated with this change of nutrition access, the host is thought to derive a benefit in the form of increased absorption efficiency (*Rawls, Samuel & Gordon, 2004*). Although there have been some studies on the gastrointestinal microbiome of sea turtles (*Ahasan et al., 2017*; *Price et al., 2017*), as far as is known, there is a lack of study relation to gut microbiome of *Trachemys scripta elegans*. Therefore, the present study analyzes, for the first time, the changes in the intestinal bacterial community of red-eared sliders during the post-hatching period via the use of a high-throughput sequencing approach. Our results provide insights into the changes in intestinal bacterial communities in the early life of this invasive turtle species, and provide basic knowledge on host-bacteria associations in turtles.

## MATERIALS AND METHODS

### Animals and sample collection

Twelve red-eared sliders were randomly selected from the Hongwang turtle farm (Hainan, China) after they were hatched immediately. The newly hatched turtle was still placed in incubator with vermiculite, until its exogenous yolk sac was assimilated almost at day 8–9 after hatched. Then, these remaining nine turtles of 10-days were mixed farmed in a cement pool with exposed tap water and fed with standard diet (turtle food, Inch-Gold, China). At 0 d, 10 d, 20 d, 30 d after hatching, these turtles were measured by vernier caliper for their body heights, carapace lengths and carapace widths, and weighed for their body weights. Then three turtles were randomly selected and anesthetized at −20 °C cryoanesthesia for 30–60 min. All turtles were euthanized by decapitation, and the intestinal tracts were collected separately and frozen immediately in liquid nitrogen. Water samples were also collected from random sites in the pool in which the turtles of 10-day, 20-day, and 30-day were living. All samples were stored at −80 °C and used for extraction of total DNA.

### DNA extraction and 16S rRNA gene sequencing

Total genomic DNA was extracted from intestine samples and environmental water samples using a Stool DNA Kit (OMEGA Bio-Tek, Norcross, GA) as per the manufacturer's instructions. Then V4–V5 region of bacterial 16S rRNA gene were amplified by using barcode primers 515F (5′-GTGCCAGCMGCCGCGGTAA-3′) and 907R (5′-CCGTCAATTCMTTTRAGTTT-3′) (*Sun et al., 2013*; *Yu, Han & Fu, 2019*). The amplicons were pooled, purified and then quantified using a Nanodrop (Thermo Scientific, MA). The purified amplicons were used for next-generation sequencing using Illumina Hiseq2500 PE250 by Sagene Biotech Inc. (Guangzhou, China).

### Data analysis

The reads archived by high-throughput sequencing were filtered to remove both those containing ambiguous nucleotides and primers sequences. The clean reads were used for further data analysis. Sequence sets showing 97% identities were defined as an Operational Taxonomic Unit (OTU) and were used for diversity (Shannon index and Simpson index) (*Simpson, 1949*), and richness (Ace index and Chao1 index) (*Chao, 1984*) analysis using QIIME (*Caporaso et al., 2010*). Rarefaction curves, alpha diversity, and beta diversity calculations were also analyzed using QIIME (*Caporaso et al., 2010*). Taxonomic assignments of OTUs that reached the 97% similarity level were made using the QIIME software package through comparison with SILVA (*Quast et al., 2013*), Greengene (*DeSantis et al., 2006*) and Ribosomal Database Project (RDP) databases (*Wang et al., 2007*). A heat map was generated using the heat map function in R (http://www.r-project.org/) with row normalized. The similarity among the microbial communities was determined using weighted UniFrac distances analysis in principal coordinate analysis (PCoA) (*Lozupone & Knight, 2005*). The linear discriminant analysis (LDA) effect size (LEfSe) method was used to identify the most differentially abundant taxons between groups, which would help discover biomarkers (*Segata et al., 2011*). Taxa were regarded as being a statistically different biomarker when the LDA scores were

**Table 1 Physiological characterization of red-eared slider turtles at different growth stages.**

| Physiological index | 0 day | 10 day | 20 day | 30 day |
|---|---|---|---|---|
| Body weight (g) | $8.25 \pm 0.55^{ax}$ | $7.53 \pm 0.58^{a}$ | $9.31 \pm 0.90^{b}$ | $10.59 \pm 1.19^{c}$ |
| Carapace length (mm) | $32.20 \pm 1.07^{a}$ | $32.20 \pm 1.15^{a}$ | $34.51 \pm 0.81^{b}$ | $37.56 \pm 0.83^{c}$ |
| Carapace width (mm) | $30.73 \pm 1.09^{a}$ | $31.30 \pm 0.96^{a}$ | $33.60 \pm 0.75^{b}$ | $36.57 \pm 0.98^{c}$ |
| Body height (mm) | $16.38 \pm 0.58^{ac}$ | $15.12 \pm 0.58^{b}$ | $15.88 \pm 0.93^{ab}$ | $17.04 \pm 0.78^{c}$ |

Notes.
$^{x}$Values sharing the same superscript letters are not significantly different from each other as determined by a multiple comparison test of least significant difference (LSD) ($p < 0.05$).

$\geq 4$. Statistical analysis was performed using ANOVA for morphological index, and Linear General Model for the diversity and richness index with initial body weight as the covariate. The value of $P < 0.05$ was considered to be statistically significant.

### Nucleotide sequence accession numbers

Raw sequences of this project were deposited in NCBI non-redundant nucleotide database under SRA accession number SRP154277 (https://doi.org/10.6084/m9.figshare.9204992. v1).

### Ethical statement

Experimental animal procedures had the prior approval of the Animal Research Ethics Committee of Hainan Provincial Education Centre for Ecology and Environment, Hainan Normal University (permit no. HNECEE-2014-004, as defined by Chinese regulations).

## RESULTS

### Turtle growth

For the first 10 days after hatching, turtles were raised without feeding, because of the existence of post-hatching yolk. During this period, the turtles showed a slight decrease in their body height. However, body weight, carapace length and width showed no significant change over this time ($p > 0.05$). However, after growing for 20 or 30 days, body weight, carapace length and width were significantly increased ($p < 0.05$) (Table 1). There were significant differences in body height between 0-day vs. 10-day, 10-day vs. 30-day, and 20-day vs. 30-day ($p < 0.05$).

### The diversity and richness index

Among the 12 samples of gastrointestinal tract, a total of 578,559 qualified reads, with an average of 48,213 reads per sample were archived (Table 2). Analysis of the qualified reads lead to the classification of 15,173 OTUs, with an average of 1,264 OTUs per sample. Both Shannon index and Simpson index showed the within-habitat diversity of samples. The Shannon index is most sensitive to changes in the importance of the rare species in the sample, while the Simpson index is most sensitive to changes in the most abundant species in the sample. Both Shannon index and Simpson index were higher at 10-day, 30-day, and water than that at 0-day and 20-day (Figs. S1A and S1B). Based on the analysis of Ace and Chao1, which represent the richness of OTU in a bacterial community, and both were higher at 10-day and 20-day than that at 0-day, 30-day, and water (Figs. S1C and S1D).

**Table 2  Bacterial richness and diversity in intestine in red-eared sliders at different growth stages and water.** Reads indicate the number of sequences archived by high-throughput sequencing. OTUs were defined at 3% dissimilarity. The richness estimators (ACE and Chao) and diversity indices (Shannon and Simpson) were calculated. Date are expressed as adjusted Mean ± SE, with body weight as the covariate. Covariates appearing in the model are evaluated at the following values: body weight = 8.92 g.

|  | Group | Reads | OTUs | Chao1 | Ace | Shannon | Simpson |
|---|---|---|---|---|---|---|---|
| Intestine | 0 day | 53419.0 ± 5781.6 | 751.7 ± 285.3[x] | 2196.6 ± 281.3[a] | 2186.5 ± 304.0[a] | 2.50 ± 0.81[a] | 0.63 ± 0.16[a] |
|  | 10 day | 41455.0 ± 12238.2 | 1532.7 ± 337.0[b] | 4974.1 ± 1118.8[b] | 5224.1 ± 1118.0[b] | 4.41 ± 0.36[b] | 0.88 ± 0.031[b] |
|  | 20 day | 52621.7 ± 5297.6 | 1511.3 ± 99.0[b] | 4935.0 ± 356.8[b] | 5190.6 ± 557.1[b] | 3.48 ± 0.37[ab] | 0.74 ± 0.58[a] |
|  | 30 day | 45357.3 ± 5474.2 | 1262.0 ± 88.5[ab] | 3475.8 ± 546.7[a] | 3444.4 ± 558.6[a] | 4.35 ± 0.15[b] | 0.88 ± 0.0058[b] |
| Environment | Water | 44824.7 ± 9255.9 | 680.7 ± 167.4 | 1533.2 ± 369.4 | 1465.5 ± 482.5 | 4.36 ± 0.74 | 0.87 ± 0.053 |

**Notes.**

[x]Values sharing the same superscript letters are not significantly different from each other according to a multiple comparison test of least significant difference (LSD) ($p < 0.05$).

To better visualize the relationships among samples, an MDS plot was generated with all replicates and sampling points clustered by development stage (Fig. 1). Based on the ordination of the weighted Unifrac distances, different gut microbiome distribution among these samples from different time. A significant difference was found between the microbiome of the water in the holding tank and turtle intestines. Moreover, microbiomes at 10-day post-hatching formed a tight cluster that did not overlap with other samples. The turtles at 0-day also displayed a significantly different microbial community although closer to those exhibited by the turtles at 10-day. The microbial communities overlapped between 20-day and 30-day but were distinct from the 0-day and 10-day.

## Abundance and significant differences between the four growth stages at the phylum level

Among these samples, the top 15 phyla were represented in Fig. 2A, and the most prominent phyla were listed in Fig. 2B and Table S1. Among the newly hatched turtles, the dominant phylum was Firmicutes (58.23%), followed by Proteobacteria (41.42%), and then Bacteroidetes (0.32%). At 10-day, the percentage of Firmicutes increased greatly to 92.94% whereas Proteobacteria decreased sharply to 5.68%, and no significant change was seen for Bacteroidetes (0.31%). At 20-day, Bacteroidetes increased strongly (27.17%) while Firmicutes (67.08%) decreased, and Proteobacteria remained relative stable (5.74%). At 30-day, the dominant phyla were Firmicutes (56.46%), Proteobacteria (20.66%), and Bacteroidetes (22.55%). Furthermore, the ratio of Firmicutes to Bacteroidetes 0-day, 10-day, 20-day, and 30-day was 181.97, 299.80, 2.47, and 2.50, respectively. In samples of environmental water, Proteobacteria (70.33%), Bacteroidetes (23.24%), Actinobacteria (4.45%), and Firmicutes (1.90%) were found to be the major bacterial phyla represented.

## Abundance and significant differences between the four growth stages at the family level

A total of 166 bacterial families were identified and the most abundant families are shown in Fig. 3A and Table S2. At 0-day, the dominant families were Clostridiaceae (13.03%), Enterobacteriaceae (23.52%), and Paenibacillaceae (36.01%). Along with the growth of the turtles, the bacteria from Paenibacillaceae sharply decreased at 10-day (0.86%) and disappeared at 20-day and 30-day. At 10-day, bacteria belonging to

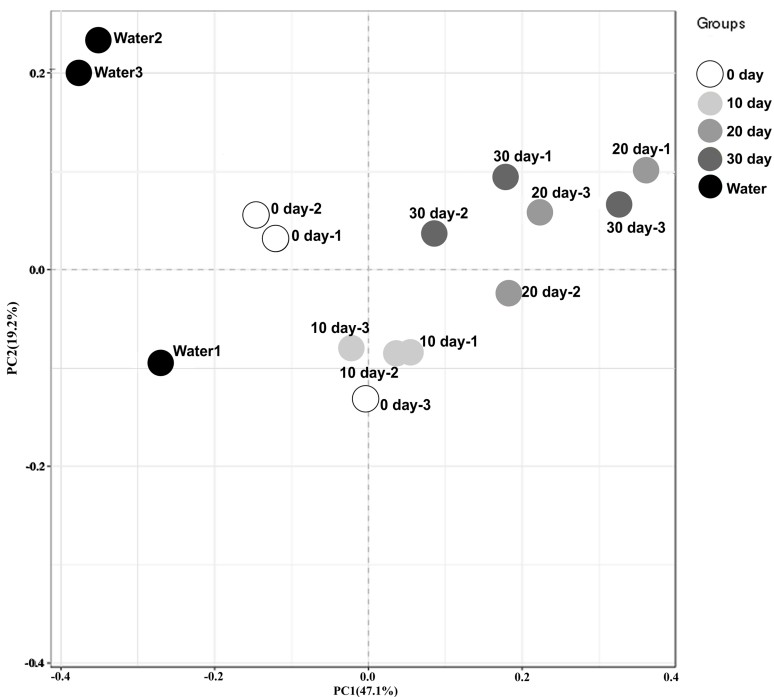

**Figure 1** **Beta diversity analysis of microbiomes from turtle intestines at different growth stages and of the environmental water ($n = 3$ samples of each condition).** Contribution of different taxonomic groups to separation of samples based on phylogenetic information. The samples were clustered by PCoA plots using weighted UniFrac distances.

the families Lachnospiraceae (33.75%), Clostridiaceae (30.24%), Peptostreptococcaceae (19.94%), Staphylococcaceae (4.07%), Moraxellaceae (3.75%) were the dominant intestinal components. Except for the Moraxellaceae family, all the other families belong to the Firmicutes phylum and made up to 92.94% of the total intestinal microbiome. At 20-day, the dominant families were Lachnospiraceae (27.74%), Bacteroidaceae (27.36%), Peptostreptococcaceae (23.43%), Clostridiaceae (9.39%), Enterobacteriaceae (5.34%), and Ruminococcaceae (3.00%). Although the Clostridiaceae increased sharply during the first 10 days, it had decreased significantly at 20-day. By contrast, Bacteroidaceae family bacteria, which were absent in 10-day turtles were a dominant group in intestine of 20-day and 30-day old turtles. At 30-day, the dominant families were Bacteroidaceae (22.56%), Clostridiaceae (18.13%), Peptostreptococcaceae (17.60%), Enterobacteriaceae (12.56%), and Lachnospiraceae (8.18%). In brief, the families Lachnospiraceae, Ruminococcaceae and Peptostreptococcaceae were also major bacterial components in intestines of turtles older than 10 days. The water microbiome showed the obviously different with the gut microbiome (Fig. 3B; Table S2). Comamonadaceae (22.86%), Aeromonadaceae (12.18%), Weeksellaceae (8.95%), Xanthomonadaceae (7.84%), Cytophagaceae (7.71%) were the dominant families in the water microbiome, while the relative of abundance of these families bacteria was very low.

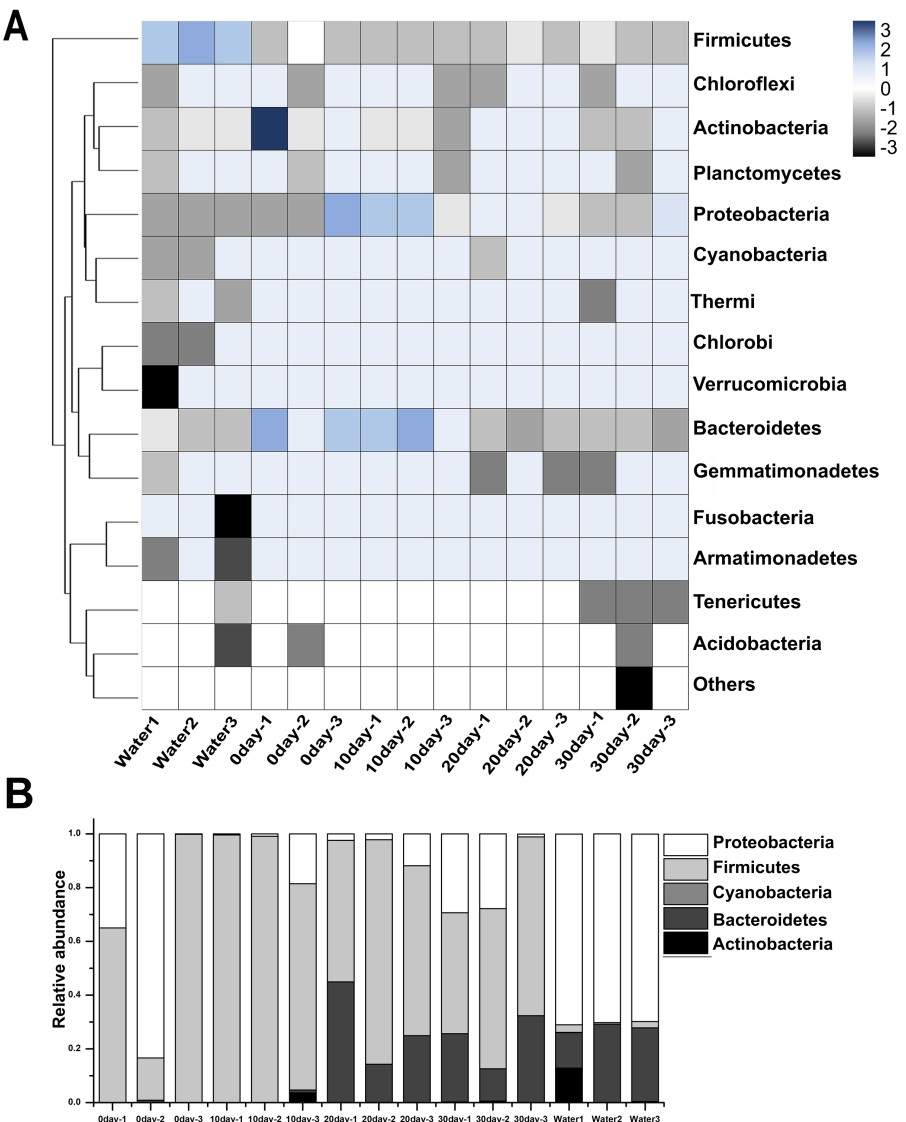

**Figure 2** **Taxonomic compositions of turtle intestinal microbiomes and water microbiomes from different samples at the phylum level.** (A) Heat map of the microbial composition of the red-eared slider gastrointestinal tract, and the water with row normalized. The heat map indicates the relative abundance of each phylum in different samples. (B) Relative abundance of sequences belonging to the dominant phyla from different samples. A color and bar plot shows the average bacterial phylum distribution in different samples. Sequences that could not be classified into any known group were assigned as 'Others'.

## Abundance and significant differences between the four growth stages at the genus level

At the genus level, the detected sequences from all samples were assigned into 323 genera. The genus *Clostridium* dominated in all the tested turtle samples, ranging from 8.23% to 24.98% (Table S3). At 0-day, the most abundant genera were *Paenibacillus* (36.01%), *Citrobacter* (20.78%), *Clostridium* (11.95%), and *Achromobacter* (10.75%). At 10-day, *Clostridium* (24.98%) and *Epulopiscium* (31.91%) were the dominant genera. The relative

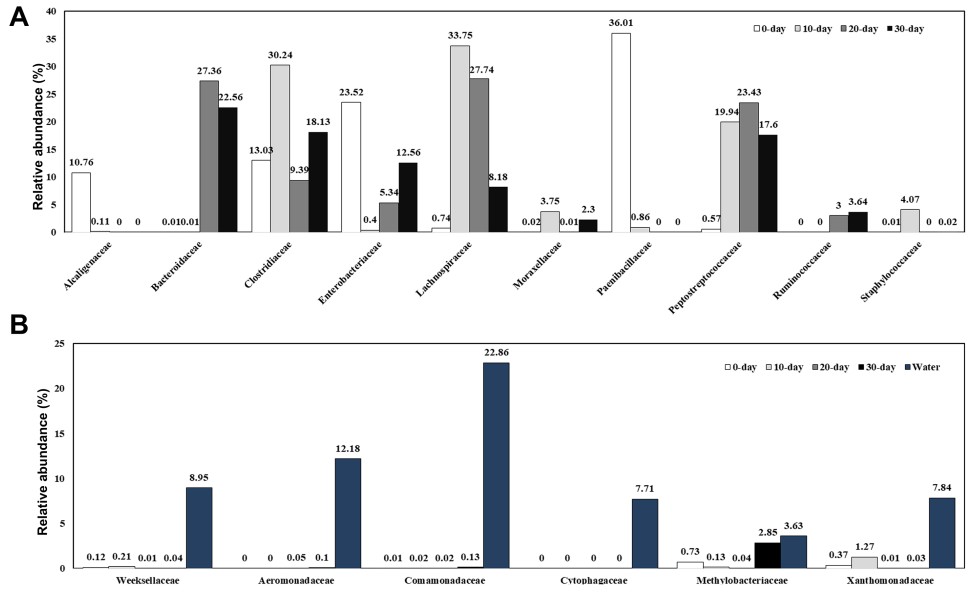

**Figure 3** Relative composition of the intestinal microbiome of red-eared sliders for different growth time (A) and the water microbiome (B) at the family level. A color bar plot shows the average bacterial family distribution in different samples.

abundance of *Epulopiscium* in 10-day turtles was 31.91%, higher than that in 0-day (0.15%), 20-day (1.42%) and 30-day (0.72%). Similarly, the relative abundance of *Clostridium* was also highest in 10-day, and decreased to 8.23% at 20-day and 17.5% at 30-day.

## Specific bacterial communities at each growth stage

The specific OTU numbers of 0-day, 10-day, 20-day, and 30-day turtles were 1299, 2845, 2509, and 1704, respectively. Among these identified OTUs, only 47 of them were shared by all turtles (Fig. 4; Table S4). These core OTUs belonged to three families, the Lachnospiraceae, Peptostreptococcaceae, and Clostridiaceae with relative abundances of 9%, 23%, and 68%, respectively. There were only 148 conserved OTUs between the intestines of 0-day and 30-day turtles, whereas 630 OTUs were conserved between the 20-day and 30-day.

To identify the specific bacteria in the intestines of turtles at different stages, biomarkers of bacteria belonging to different taxonomic groups were analyzed (Fig. S2; Fig. 5; Table S5). At family level, the significant different families were Paenibacillaceae and Alcaligenaceae at 0-day, Lachnospiraceae at 10-day, Bacteroidaceae and Odoribacteraceae at 20-day, and Erysipelotrichaceae at 30-day. At genus level, *Paenibacillus* and *Achromobacter*, *Epulopiscium*, *Blautia* showed the significant difference in the 0-day, 10-day and 30-day turtles, respectively, whereas no genus reached a taxonomic level score (LDA score ≥4) in 20-day turtles. In summary, these different taxa could be used as distinguishing biomarkers. The healthy red-eared slider revealed 10 such microbial biomarkers that showed significant differences in abundance at different growth stages in the standardized condition without considering other factors.

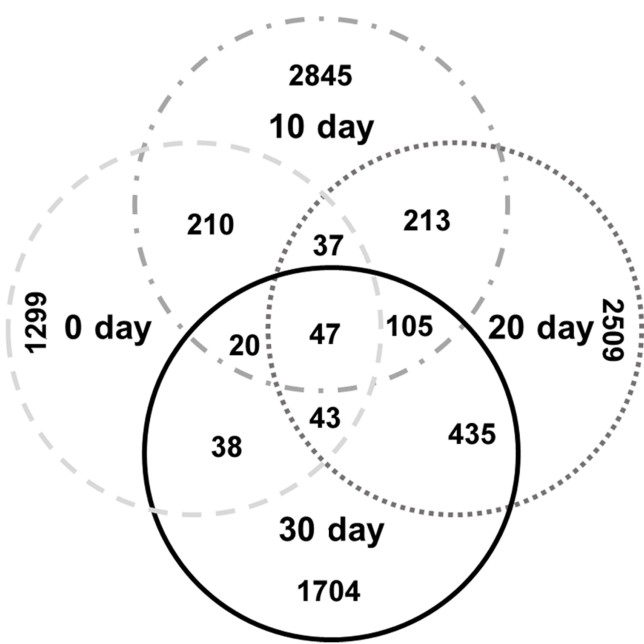

**Figure 4 Analysis of the differences in the intestinal microbiome among different growth stages in turtles.** Venn diagrams showing the number of shared and exclusive OTUs in different growth stage turtles.

## DISCUSSION

In this study, the first detailed analysis of the intestinal bacterial communities of the red-eared slider turtle over the first 30 days post-hatching was performed using a high-throughput sequencing approach. During the first 9 days post-hatch, the turtles were fasting and acquired nutrition from the yolk sac and, hence, did not need to digest and assimilate nutrition via the intestine. Therefore, the richness of the intestinal microbiome was the lowest at 0-day and growth before the first feeding was very slow. After the closure of their umbilical fontanel, the turtles were fed with a commercial diet. In our study, the 10-day turtles showed the highest intestinal microbiome diversity and abundance. Very short dietary manipulations can have a rapid and substantial impact on the intestinal microbiome (*David et al., 2014*). This was consistent with other researches that diet could affect intestinal microbial diversity (*Claesson et al., 2012*; *Scott et al., 2013*). The results suggested that the microbiome of the turtle adjusts itself to optimize the bacterial species needing for the particular food that is given which is in this case is a constant single food. Another explanation was the richness and diversity of gut microbes varied with development stage in turtles. This is consistent with previous reports showing that the diversity and richness of gut microbiome of pigs and chicks decreased with age (*Ballou et al., 2016*; *Zhao et al., 2015*). For the 30-days turtles, the intestinal microbiomes diversity increased again (Table 2), which endowed the turtles with ability to face to different diet type and enhanced their tolerance to the environment.

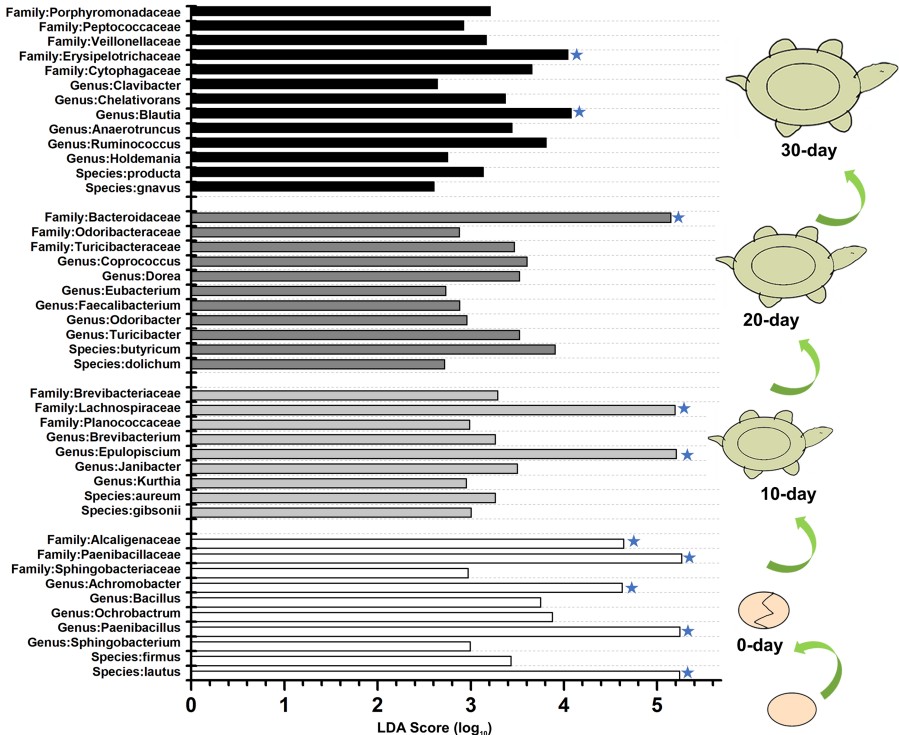

**Figure 5** **LEfSe identified the most differentially abundant bacterial taxa at different growth stages of turtles at different taxonomic levels.** In the figure, 0 day, 10 day, 20 day, and 30 day indicate the intestinal microbiome of turtles growing at 0 days, 10 days, 20 days, and 30 days. Only taxa meeting an LDA significant threshold >2 are shown. The taxa could be regarded as a biomarker when LDA significant threshold >4, and was marked by a "star". The eggs and turtles drawings in the figure are original and drawn by Qin Peng.

The results revealed that the gut microbiome of the turtle exhibited temporal differences in composition at the genus level and beyond. In the intestinal microbiome of 10-day turtles, bacteria from the *Epulopiscium* genus showed the highest relative abundance. *Epulopiscium* is a group of giant bacteria that was reported to be widely distributed in the intestinal tracts of herbivorous surgeonfish with high abundance (*Miyake, Ngugi & Stingl, 2016*). The predicted function of *Epulopiscium* is to facilitate the digestion and decomposition of ingested food (*Flint et al., 2008*; *Thomas et al., 2011*). These results suggested that *Epulopiscium* bacteria might facilitate the turtle to access nutrition from the diet. After the fasting time (during yolk consumption), the turtles were fed with commercial food, which lead to changes in the intestinal microbiome composition, and formed intestinal microbiota that were suitable for digesting the commercial diet.

At family level, the Bacteroidaceae, Lachnospiraceae, and Peptostreptococcaceae were the dominant bacteria in the turtles after growing for 10 days. Bacteria of these families are responsible for metabolizing carbohydrate complexes, such as cellulose (*Price et al., 2017*), and degrading plant material to maintain gut health in several animals (*Biddle et al., 2013*). These dominant families were perhaps a good preparation by the gut for the foods that

the juvenile turtles will begin to ingest, which was fitting well with the starvation tolerance behavior of this kind of turtle.

Many studies have shown that Bacteroidetes, Firmicutes and Proteobacteria are numerically dominant phyla in the gut microbiome of animals (*Thomas et al., 2011*), including humans (*Qin et al., 2010*), wild-captured green turtles (*Chelonia mydas*) (*Ahasan et al., 2017*), loggerhead turtles (*Caretta caretta*) (*Abdelrhman et al., 2016*), doves, waterfowls and geese (*Hird et al., 2015*; *Li et al., 2017*) and other vertebrates (*Ley et al., 2008*). In the present study, although Proteobacteria made up a large proportion of the sequences in the intestinal microbiome of 0-day turtles, overall, the Firmicutes and Bacteroidetes were the most ubiquitous and common, which is consistent with previous reports. The relative abundance of Bacteroidetes phylum was extremely low during the first 10 days, and then rapidly increased to a dominant phylum at 20-day and 30-day. Bacteroidetes is a degrader of polymeric organic matter to help the host digest food. The interaction between Bacteroidetes and the host is known to be mutualism since the fitness of both partners is increased (*Backhed et al., 2005*). Bacteroidetes also function to activate T-cell mediated responses (*Mazmanian, 2008*), limit the colonization of the potential pathogens (*Mazmanian, Round & Kasper, 2008*), and produce antineoplastic properties (*Thomas et al., 2011*). Bacteroidetes can colonize all parts of gastrointestinal tract, despite of the differences in ambient pH, nutrients, and oxygen availability in different parts (*Thomas et al., 2011*). So, the increase of Bacteroidetes with the turtles grows may endow the host with stronger ability to digest food and resist disease. Species of the Firmicutes phylum can process food with a higher abundance of insoluble carbohydrate (*Jandhyala et al., 2015*). A recent study showed that the ratio of Firmicutes to Bacteroidetes was of significant relevance in signaling human gut microbiota status (*Mariat et al., 2009*), the relative proportion of Bacteroidetes decreasing in unhealthy humans (*Ley et al., 2006*). Another research revealed higher relative abundance of Bacteroidetes could help keep the turtle healthy (*Ahasan et al., 2018*). In our study, the ratio of Firmicutes to Bacteroidetes at 0-day and 10-day was much higher than at 20-day and 30-day, which indicated that it tended to be stable in the older turtles, and the relative proportion of Bacteroidetes increased. So, a stable composition of Firmicutes and Bacteroidetes and a relatively higher proportion of Bacteroidetes could maintain intestinal stability to make the turtle healthier, and allow it to adapt to environmental surroundings and food availability more quickly. Several factors, including host intestinal structure, physiological status, habitat, developmental stages, feeding strategy, living environment, and certain environment conditions, were reported to affect the composition of the host intestinal microbiome (*Ahasan et al., 2017*; *Ravussin et al., 2012*). Nevertheless, in the present study, there were no relationship between the water microbiome and the dominant phyla Proteobacteria and Firmicutes of the turtle intestine. Thus, the intestinal bacteria of these phyla were rarely affected by living water environment without considering other factors. So, it can be inferred that the intestinal microbiomes of red-eared slider may be self-regulating, which may endow the turtles with the ability to readily adapt into different environments, different foods and seasonal changes. The Proteobacteria phylum was rich in the water environment, but in the intestinal

microenvironment, the relative abundance of Proteobacteria was low in 10-day and 20-day turtles, but increased in the 30-day turtle gut. A previous study reported that after Bacteroidetes break down proteins into amino acids, and chitin into N-acetylglucosamine, Proteobacteria could take up and process in these monomers (*Cottrell & Kirchman, 2000*). Thus, Proteobacteria could help the turtle to get nutrition from degraded monomers. Overall, the dominant existence of Proteobacteria, Firmicutes and Bacteroidetes could be one reason that turtles are well-adapted to a flexible environment. In the wild, food types would vary on a daily or seasonal basis and also with the specific environment, such as lake, river, brackish coastal water, etc. In our study, the living water environment does not affect the gut microbiome of red-eared slider under the standard condition. Based on the literature available in humans and other animal species (*Chen, He & Huang, 2014*; *Zha et al., 2018*), it is possible that multiple exogenous factor, including food or feed, influences the intestinal microbiota community.

In the family and below level, total 10 biomarkers were discovered in the 0-day, 10-day, 20-day and 30-day samples. The biomarkers were obviously different for the different age groups, so it inferred that gut biomarker might help to predict age of this turtle.

## CONCLUSIONS

The diversity and richness of the intestinal microbiomes were the lowest at 0-day, and highest at 10-day because of its first offer with food. With the turtles growing, the high abundance of Firmicutes, Bacteroidetes, Proteobacteria and the stable ratio of Firmicutes to Bacteroidetes could help maintain intestinal stability to make the red-eared slider healthier, and endow it with high adaptive capacity for dealing with different environments. Peptostreptococcaceae, Lachnospiraceae and Clostridiaceae were core bacterial communities at all growth stages in the red-eared slider. At the class level, Bacilli, Clostridia, Bacteroidia and Erysipelotrichi could be regarded as distinguishing biomarkers in the intestine of 0-day, 10-day, 20-day and 30-day turtles, respectively.

### Funding

This work was supported by the Hainan Natural Science Foundation (No. 2019CXTD404), the national Natural Science Foundation of China (No. 31760116),the National Natural Science Foundation of China (No. 31800157), and the Hainan Natural Science Foundation (No. 319QN212). The funders had no role in study design, data collection and analysis, decision to publish, or preparation of the manuscript.

### Grant Disclosures

The following grant information was disclosed by the authors:
Hainan Natural Science Foundation: 2019CXTD404, 319QN212.
National Natural Science Foundation of China: 31760116, 31800157.

## Competing Interests
Kenneth B. Storey is an Academic Editor for PeerJ.

## Author Contributions
- Qin Peng conceived and designed the experiments, performed the experiments, analyzed the data, prepared figures and/or tables, authored or reviewed drafts of the paper, and approved the final draft.
- Yahui Chen, Zimiao Zhao and Peiyu Yan performed the experiments, prepared figures and/or tables, and approved the final draft.
- Li Ding performed the experiments, authored or reviewed drafts of the paper, and approved the final draft.
- Kenneth B. Storey and Haitao Shi analyzed the data, authored or reviewed drafts of the paper, and approved the final draft.
- Meiling Hong conceived and designed the experiments, analyzed the data, authored or reviewed drafts of the paper, and approved the final draft.

## Animal Ethics
The following information was supplied relating to ethical approvals (i.e., approving body and any reference numbers):

Experimental animal procedures had the prior approval of the Animal Research Ethics Committee of Hainan Provincial Education Centre for Ecology and Environment, Hainan Normal University (permit no. HNECEE-2014-004, as defined by Chinese regulations).

## Data Availability
The sequences are available at GenBank: SRP154277.

## Supplemental Information
Supplemental information for this article can be found online at http://dx.doi.org/10.7717/peerj.8501#supplemental-information.

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
