# Peer review of "Early-life intestinal microbiome in Trachemys scripta elegans analyzed using 16S rRNA sequencing"

_PeerJ, doi:10.7717/peerj.8501_

## Round 0.1 · original submission · Major Revisions

Please make any suggested changes to the manuscript and send to us for re-evaluation.

Reviewer 1 ·

Basic reporting

The authors assessed the intestinal microbiome of red-eared slider hatchlings (fed on commercial particle food) was systematically analyzed at four different growth stages (0d,10d, 20d, 30d) by 16S rRNA sequencing approach, and discovered that the gut microbial diversity was different at different development stage, and The diversity and richness of the intestinal microbiomes were lowest at 0-day, and increased with turtle growth except for the period at the start of feeding.

Experimental design

Some conclusion is more convincing the authors can compare the intestinal microbiome of red-eared slider samples from manual environment with that from natural environment.

Validity of the findings

that's ok

Additional comments

Early-life intestinal microbiome in Trachemys scripta elegans analyzed using 16S rRNA sequencing
By Peng et al.

Review:

The authors assessed the intestinal microbiome of red-eared slider hatchlings (fed on commercial particle food) was systematically analyzed at four different growth stages (0d,10d, 20d, 30d) by 16S rRNA sequencing approach, and discovered that the gut microbial diversity was different at different development stage, and The diversity and richness of the intestinal microbiomes were lowest at 0-day, and increased with turtle growth except for the period at the start of feeding. I have some concerns as listed below. I suggest the authors to carefully revise their manuscript before further process.
Major concerns:
1. The sample size is small and I found there are different among the 3 samples from each stage from PCoA plots in figure 1 and Taxonomic compositions in figure 2, I don’t know the authors how to resolve the problems ? Moreover, why you take samples from only the 4 development stage (0,10,20,30 days after hatch)?
2. As we known, Sequence quality can affect the results, in the MS, I have not found the content about the sequence results, specially about sequencing depth.
3. The authors also sequenced the Water samples in the pool in which the turtles were living, why not compare the results with the samples at 0day? If the intestinal microbiome at 0 day is related with the water?

Other concerns:
1. Some conclusion is more convincing the authors can compare the intestinal microbiome of red-eared slider samples from manual environment with that from natural environment.

Reviewer 2 ·

Basic reporting

This paper report the analysis of the intestinal microbiota in hatchling Trachemys scripta ranging from 0 to 30 days of life.
The manuscript is overall clear and the English is good.
Provided literature and background are sufficient to understand the subject, still some personal background knowledge is needed, as for all the papers.
The work is well organized and provided raw datas are enough to understand the experimental design and results.
Results are significant and self-contained, but some generalizations made by the authors in the discussion and conclusions sections need to be revised.
Here a list of suggested revisions and some comments. The list is organized by lines (top to bottom):
Lines 25-26: The abundance of Clostridium also showed the highest value in 10-day turtles and then decreased over time.
Based on table S3 information, looks like only at 20-days you detected a decrease in Clostrium concentration.
Clostridium 11.95 24.98 8.23 17.50 0.15

INTRODUCTION:
Line 40: improving host performance
I suggest removing this sentence or to explain the meaning of “improving performance”.
Line 40: reducing environmental pollution
As stated by different papers in the recent literature, microbiome is involved in different biochemical process ending in bioactivation of drug and pollutant compounds. Furthermore, the bacterial population concentration is itself influenced by the presence of pollutant and chemicals, generally speaking. For all these reasons I would recommend to remove this sentence.
Here listed some examples:
Jin Y, Wu S, Zeng Z, Fu Z. Effects of environmental pollutants on gut microbiota. Environ Pollut. 2017 Mar;222:1-9. doi: 10.1016/j.envpol.2016.11.045. Epub 2017 Jan 11.

Yvonne Vallès and M. Pilar Francino. Air Pollution, Early Life Microbiome, and Development. Curr Environ Health Rep. 2018; 5(4): 512–521.

Yuanxiang Jin, Sisheng Wu, Zhaoyang Zeng, Zhengwei Fu. Effects of environmental pollutants on gut microbiota. Environmental Pollution. Volume 222, March 2017, Pages 1-9.
Lines 44-45: colonization resistance
I would suggest authors to include this paper in the reference list.
Ducarmon QR, Zwittink RD, Hornung BVH, van Schaik W, Young VB Kuijper EJ. Gut Microbiota and Colonization Resistance against Bacterial Enteric Infection. Microbiol Mol Biol Rev. 2019 Jun 5;83(3). pii: e00007-19. doi: 10.1128/MMBR.00007-19. Print 2019 Aug 21.

Line 48: between host health, immunity and disease resistance.
Please consider placing a comma (,), before the and.

Line 59: …invasive environment (Ma et al. 2013; Zhao et al. 2013)Several of these advantages…
I am not sure what authors mean with “invasive environment”, can you please substantiate the term? Additionally, place a full stop (.) and a space, after citation brackets.
Line 70: (absorbed into the abdominal cavity)
Replace abdominal cavity, with celomatic cavity.
Lines 70-71: In captive animals the diet is primarily commercial particle food that has high levels of protein.
Place a reference, please.
Lines 71-72: By selecting for bacteria associated with this change of nutrition access, the host is thought to derive a benefit in the form of increased absorption efficiency.
Place a reference, please. Furthermore, I am not sure how commercial diets could have had an influence on the biological success of Trachemys spp.
Line 73: The present study analyzes (for the first time) the changes in the intestinal bacterial community…
Please, consider replacing brackets with comma (,).
Line 80: Newly hatched red-eared sliders were randomly selected from the Hongwang turtle farm …
It would be thoughtful of you to tell readers how much time after hatching the turtles were collected. If it is immediately after hatching, please say so. Additionally, please clearly state the number of animals included in the study.
Lines: 82-84: The external yolk sac of newly hatched turtles was typically almost fully assimilated by day 10 after hatching and so a standard diet (turtle food, Inch-Gold, China) was first fed after day.
I suppose that yolk sac resorption timing was an observation you made; in this case, consider writing the exact number of days in the result section and here just place a generic “after yolk sac assimilation”. Additionally, “fully assimilated” might be misleading because after closure of the umbilical fontanel, this structure is still present in the celomic cavity (as you stated in the introduction). So, I suggest replacing the sentence with:
After complete closure of the plastron umbilical fontanel, the animals were feeded with standard diet (turtle food, Inch-Gold, China).
Line 84: physiological indexes
Can you please specify what do you mean with this term?
Lines 99-100: The reads archived by high-throughput sequencing were filtered to remove reads containing ambiguous nucleotides and the reads of the primers, and the clean reads were used for further data analysis.
The extensive repletion of the word “reads” makes the sentence unclear. Since I agree that “reads” is the proper terminology, I suggest splitting the sentence.
The reads archived by high-throughput sequencing were filtered to remove both those containing ambiguous nucleotides and primers sequences. The clean reads were used for further data analysis.
Line 106: with SILVA(Quast et al. 2013),
Place a space before the bracket.
Line 125: cultured
Please, replace cultured with raised.
Line 128: body weight, body height, carapace length and width
The procedure you used to perform these measurements must be included in the materials and methods section.
Line 141: difference
Is this different?
Line 138: richness
I am sorry, with richness you mean the overall concertation of bacteria or the number of bacterial populations, or both? Please, consider to rephrase the sentence in a cleared way.
Lines 139-140: highest intestinal abundance after growing for 10 and 20 days
The overall concentration was higher at 10 or 20 days? If was equal, please, state it.
Line 157: was 181.97, 299.80, 2.47, 2.50.
Place an and in front of 2.50.
Lines 180-181: The abundance of Clostridium was also highest in 10-day turtles but decreased somewhat as turtles transitioned to eating commercial food.
I suggest removing this sentence, because as is it is not informative and not even, somewhat, correct.
Line 183: The specific OTU numbers of 0-day, 10-day, 20-day, and 30-day turtles were 1299, 2845, 2509 and
Place a comma (,) before the and.
Lines 185-186: These core OTUs were belonged to three families, the Peptostreptococcaceae, Lachnospiraceae and Clostridiaceae with relative abundances of 23%, 9% and 68%, respectively.
It would be nice to have bacteria listed in order of increasing relative abundance.
Line 192: Bacteroidia and Erysipelotrichi
Consider placing a comma (,) before the and.
Lines 197-199: In summary, these different taxa could be used as distinguishing biomarkers. The red-eared slider revealed 20 such microbial biomarkers that showed significant differences in abundance at different growth stages.
I have a doubt about this sentence. You substantiated that environmental water microbiota dose not seems to affect intestinal microbiota, but what about food and other environmental conditions, such as temperature and season? What about diseased animals? Can you be 100% sure that these bacterial population can be used as a generalized biomarker?
Line 244: produce antineoplastic properties
I am sorry, do you mean that produce antineoplastic substances, or that has anti-neoplastic properties?
Line 260: Thus, the intestinal bacteria of these phyla were rarely affected by environment.
I would use caution saying something like that. Your experimental conditions were standardized, and you might have had no chance of observing environmental effects on the intestinal microbiota.
Line 260-262: So, it could be deduced that the intestinal microbiomes of red-eared slider were mainly regulated by itself, which would endow the turtles with the ability to readily adapt into different environments, different foods and seasonal changes.
You might be right, but you cannot prove it. Please, rephrase the sentence in a more dubitative way.
Lines 271-272: It could lead to adjustments to the microbiome populations, whereas in the captive environment, almost no changes in commercial food might stabilize the bacterial populations more quickly.
This sentence underling that environment can affect intestinal microbiota, and this is quiet in disagreement with what have been wrote above.

Table S1: 20-dayd.
Replaced with 20-day. Please apply the change to all the other tables.
Figure1: Please, consider to maker the legend more readable. Remove TSE and just put Day 0, Day 10,… What does TSEe mean? Which type of grouping represent? It is the water?
Figure 4b: Can you please consider to put all the letters in black and bolted?
Figure 5: In the figure legend, state the meaning of the “star” placed near to some of the bacteria names.
Table S6: Change “Number of sequence” to Sequence number, if I correctly understood what you mean.

Experimental design

This work is well designed and structured but, as many other in the filed, has a major limit, that is lost of variety. I wish to point out, that if you take 12 animals, from the same breeding center, you keep them together (even if in small groups), with the same diet, very limited environmental variables, and a single measure for each animals, generalizations and comparison with wildlife condition must be expressed very carefully.
Some information is missing in the material and methods, such as how you performed euthanasia and how did you take turtles measurements.
Additional suggestion are reported in the "basic reporting" part.

Validity of the findings

The work is innovative and provide some clue on the ecological success of Trachemys spp.
Data are provided and statistical and bioinformatics analysis look correct.
Conclusions and discussion need minor revision in the view to reduce inferences on the finding applied to less controlled conditions.

Additional comments

It would be nice to expand your research by adding wild caught animals and increasing variability. Additionally, you might think to perform cloacal swabs instead of killing the animals; this will allow you to repeated analysis on the same animal. In this way you may take some hatchling Trachemys and "follow" their microbiota evolution over a longer period of time.

Reviewer 3 ·

Basic reporting

The manuscript is generally clear and well-written, although there are several instances in the manuscript that could benefit from more clear explanation and specific details. For example, it is not clear when the water samples were collected (i.e., at which time point). Also, the Methods suggest that feeding began after the 10-day time point, but in the discussion of the results it less clear whether turtles sampled at 10 days had access to food prior to euthanasia or not (see attached pdf for more specific comments). Good background material is provided, but there is very little information about what is currently known about the gut microbiome in other turtle species. Such information would be helpful for providing comprehensive context and relevance of the current study. The manuscript is structured appropriately.

Experimental design

Study conforms to the aims and scope of the journal and the research question is well defined – it is a descriptive survey of the gut microbiome in developing red-eared slider turtles. I see no obvious issues with the study design, but the limited scoped does restrict the types of conclusions that can be drawn. Methods are generally good, but there are some important pieces that should be explained in more detail, such as whether the alpha-diversity measures used for analyses were produced by Mothur or Qiime (see attached pdf for more specific comments).

Validity of the findings

Data are provided. The results of this study are interesting because they provide baseline information about the gut microbiome in this species and because they document microbiome changes over early turtle development. However, the relatively small sample size and comparative nature of this study do not allow for the types of strong conclusions that the authors propose in the Discussion section regarding adaptability and nutritional efficiency. A general suggestion to the authors would be to use figures to illustrate changes in relative abundance of selected bacterial taxa exhibiting significant variation with age, including appropriate statistical results. Presenting these data in figure form as opposed to in-text would help the reader to better interpret the nature and relevance of these changes. I have included specific comments on which speculations in the Discussion could benefit from clarification and better support from the literature (see attached pdf for more specific comments).

Additional comments

Please see attached pdf for both general and specific comments on this manuscript

Annotated reviews are not available for download in order to protect the identity of reviewers who chose to remain anonymous.

---

## Round 0.2 · Minor Revisions

Please answer the questions from the reviewer and send a rebuttal letter with these.

Reviewer 2 ·

Basic reporting

Line 87: ability to exploit new food supplies in a new environment
Consider to rephrase in something like: “… ability to adapt to different environment by exploit new food sources/supplies”

Lines 89-100: The community of the gut microbiome varies at different growth stages (Burgos et al. 2018; Dulski et al. 2018; Huang et al. 2014).
Can you find some references from other turtles/tortoises? All the reported references are from fishes.

Lines 100-105: Although there have been some studies on the gastrointestinal microbiome of sea turtles (Ahasan et al. 2017; Price et al. 2017), as far as is known, there is a lack of studies relation to gut microbiome of Trachemys scripta elegans
There is a project on the intestinal metagenome in Trachemys. See the link below.
https://www.ncbi.nlm.nih.gov/bioproject/481331

Lines 116-117: The newly hatched turtle was still placed in incubator with vermiculite, until its exogenous yolk sac was assimilated almost at day 5-10 after hatched
And in account to wat stated in the rebuttal:
Lines: 82-84: The external yolk sac of newly hatched turtles was typically almost fully assimilated by day 10 after hatching and so a standard diet (turtle food, Inch-Gold, China) was first fed after day.
I suppose that yolk sac resorption timing was an observation you made; in this case, consider writing the exact number of days in the result section and here just place a generic “after yolk sac assimilation”. Additionally, “fully assimilated” might be misleading because after closure of the umbilical fontanel, this structure is still present in the celomic cavity (as you stated in the introduction). So, I suggest replacing the sentence with:
After complete closure of the plastron umbilical fontanel, the animals were feeded with standard diet (turtle food, Inch-Gold, China).
Answer: This sentence has been corrected. It is difficult to give the exact number of days of yolk sac assimilated, because the days of yolk sac assimilated are different for different individuals (most of them are 10 days or so).
Please, rephrase the sentence because it is not clear and probably some words are missing. I would suggest to you not to use the word “exogenous”, and replace it with “extra celomic”. You can consider something: “Newly hatched turtles were kept in the incubator up to 5-10 days post hatching, when vast majority of the yolk was absorbed”. When did you consider the hatchlings ready to come out the incubator? Which was the size of the yolk sac (the mean value)? In the rebuttal you state that most of the animals requires 10 days or so to absorb the yolk-sac, so why you state that the incubation range time is 5-10 days? Five days of difference is a huge amount of time.


Line 118: Then, these turtles of 10-day were mixed farmed
This is not completely true, since you sacrificed 3 animals at day 0, meaning that only 9 animals underwent commercial feeding.

Line 120: after feeding, growth indexes
Do you mean after hatching?

Line 122: and anesthetized at -20℃ crymoanesthesia for 0.5-1 h
Do you mean cryoanesthesia? Please, also consider to place the timing in minutes (30-60 minutes).
I am wondering if you can consider this anaesthesia only. It is likely to me that an animal place at -20C for an hour it is likely to have permanent neural damage, especially in heliothermic species as reptiles. How did you kill the animal before removing the intestinal tract?

Lines 327-328: So, it can be inferred that the intestinal microbiomes of red-eared slider may be regulated by itself,
What about replacing “regulated by itself” with self-regulating

Lines 338-340: In our study, the living water environment can not affect the gut microbiome of red-eared slider under the standard condition.
Sorry, can not or do not?

Lines 340-342: According to other paper about the effects of food on people and other animal (Chen et al. 2014; Zha et al. 2018), so we infer that other factors such as food may affect the gut microbiome of it.
Please, consider to rephrase the sentence in something like: “Based on the literature available in humans and other animal species (Chen et al. 2014; Zha et al. 2018), it is possible that multiple exogenous factor, including food/feed, influcences the intestinal microbiota community”.

Lines 343-345: In the family and below level, total 10 biomarker was discovered in the 0-day, 10-day, 20-day and 30-day samples. The biomarker was obviously different for the different age groups, so it inferred that gut biomarker might help to predict age of this turtle.
Biomarkers? (in both the sentences).

Line 349: highest at 10-day because of its first offer with food and the change of living environment.
Based on this sentence, I understand that environmental change (i.e. Moving from the incubator to the water) has influenced the microbiota and this is not in agreement with what you stated above. Please, consider to rephrase the sentence.

Experimental design

This work is well designed and structured but, as many other in the filed, has a major limit, that is lost of variety. I wish to point out, that if you take 12 animals, from the same breeding center, you keep them together (even if in small groups), with the same diet, very limited environmental variables, and a single measure for each animals, generalizations and comparison with wildlife condition must be expressed very carefully.
Some information is missing in the material and methods, such as how you performed euthanasia and how did you take turtles measurements.

Validity of the findings

The work is innovative and provide some clue on the ecological success of Trachemys spp. Data are provided and statistical and bioinformatics analysis look correct.
Conclusions and discussion need minor revision in the view to reduce inferences on the finding applied to less controlled conditions.

Additional comments

Thank you for the answer. In my opinion would be of more interest having the follow-up from a single subject at different timepoints, compared to having multiple subjects. For example, having a single subject and see how microbiota change secondary to stress and more importantly how long dose it takes to return to normality (if ever) would be a more useful information (for example for a veterinarian). I do not know how much material you need for the microbiome, but you can also consider to perform intestinal FNA (Fine needle aspirates) to collect fecal material.

---

## Round 0.3 · Minor Revisions

I believe that you simply need to provide the Rarefaction curves, as requested by R1

Reviewer 1 ·

Basic reporting

The authors assessed the intestinal microbiome of red-eared slider hatchlings (fed on commercial particle food) at four different growth stages (0d,10d, 20d, 30d) by 16S rRNA sequencing approach, and discovered that the gut microbial diversity was different at different development stage.

Experimental design

It's ok.

Validity of the findings

Need to supply the Rarefaction curves for all samples.

Additional comments

The authors just answer my reviews in their rebuttla letter, and some I don't agree. Even you sequenced 16S rRNA, you still can check if most of the diversity presented in your samples according to Rarefaction curves, and you still not compare your results with the ones from the environment.

Reviewer 2 ·

Basic reporting

The paper is now ready to be published, in my opinion.

Experimental design

The experimental design is correct

Validity of the findings

The paper presents scientifically valid findings.

Additional comments

I would like to thank the authors for your collaboration in reviewing the manuscript.

---

## Round 0.4 · accepted · Accept

Congratulations the manuscript has been accepted and will soon be published